# Ergosta-7,9(11),22-trien-3β-ol Alleviates Intracerebral Hemorrhage-Induced Brain Injury and BV-2 Microglial Activation

**DOI:** 10.3390/molecules26102970

**Published:** 2021-05-17

**Authors:** Po-Jen Hsueh, Mong-Heng Wang, Che-Jen Hsiao, Chih-Kuang Chen, Fan-Li Lin, Shu-Hsien Huang, Jing-Lun Yen, Ping-Huei Tsai, Yueh-Hsiung Kuo, George Hsiao

**Affiliations:** 1Graduate Institute of Medical Sciences and Department of Pharmacology, School of Medicine, College of Medicine, Taipei Medical University, Taipei 11031, Taiwan; b8501092@tmu.edu.tw (P.-J.H.); hsiaocj@tmu.edu.tw (C.-J.H.); jaxonhuang0902@gmail.com (S.-H.H.); m120102039@tmu.edu.tw (J.-L.Y.); 2Department of Physiology, Medical College of Georgia, Augusta University, GA 30912, USA; mwang@augusta.edu; 3Laboratory of Neural Repair, Department of Medical Research, China Medical University Hospital, Taichung 40402, Taiwan; 4Graduate Institute of Clinical Medicine, College of Medicine, Taipei Medical University, Taipei 11031, Taiwan; leonard@cgmh.org.tw; 5Department of Physical Medicine and Rehabilitation, Chang Gung Memorial Hospital at Tayouan, Taoyuan 33378, Taiwan; 6School of Medicine, Chang Gung University, Taoyuan 33302, Taiwan; 7Menzies Institute for Medical Research, University of Tasmania, Hobart 7000, Tasmania, Australia; fanli.lin@utas.edu.au; 8Translational Imaging Research Center, College of Medicine, Taipei Medical University, Taipei 11031, Taiwan; phtsai04@csmu.edu.tw; 9Department of Medical Imaging and Radiological Sciences, Chung Shang Medical University, Taichung 40201, Taiwan; 10Department of Chinese Pharmaceutical Sciences and Chinese Medicine Resources, China Medical University, Taichung 40402, Taiwan; 11Department of Biotechnology, Asia University, Taichung 40402, Taiwan; 12Chinese Medicine Research Center, China Medical University, Taichung 404, Taiwan

**Keywords:** ergosta-7,9(11),22-trien-3β-ol, intracerebral hemorrhage, COX-2, MMP-9, microglia, JNK

## Abstract

Intracerebral hemorrhage (ICH) is a devastating neurological disorder characterized by an exacerbation of neuroinflammation and neuronal injury, for which few effective therapies are available at present. Inhibition of excessive neuroglial activation has been reported to alleviate ICH-related brain injuries. In the present study, the anti-ICH activity and microglial mechanism of ergosta-7,9(11),22-trien-3β-ol (EK100), a bioactive ingredient from Asian medicinal herb *Antrodia camphorate*, were evaluated. Post-treatment of EK100 significantly attenuated neurobehavioral deficit and MRI-related brain lesion in the mice model of collagenase-induced ICH. Additionally, EK100 alleviated the inducible expression of cyclooxygenase (COX)-2 and the activity of matrix metalloproteinase (MMP)-9 in the ipsilateral brain regions. Consistently, it was shown that EK100 concentration-dependently inhibited the expression of COX-2 protein in Toll-like receptor (TLR)-4 activator lipopolysaccharide (LPS)-activated microglial BV-2 and primary microglial cells. Furthermore, the production of microglial prostaglandin E_2_ and reactive oxygen species were attenuated by EK100. EK100 also attenuated the induction of astrocytic MMP-9 activation. Among several signaling pathways, EK100 significantly and concentration-dependently inhibited activation of c-Jun N-terminal kinase (JNK) MAPK in LPS-activated microglial BV-2 cells. Consistently, ipsilateral JNK activation was markedly inhibited by post-ICH-treated EK100 in vivo. In conclusion, EK100 exerted the inhibitory actions on microglial JNK activation, and attenuated brain COX-2 expression, MMP-9 activation, and brain injuries in the mice ICH model. Thus, EK100 may be proposed and employed as a potential therapeutic agent for ICH.

## 1. Introduction

Intracerebral hemorrhage (ICH) accounts for approximately 15% of all strokes in the Caucasian population and up to 30% of all strokes in the Asian population, despite being associated with a disproportionate level of patient deaths [1,2]. ICH also caused patients with greater worldwide disability-adjusted life years lost [3]. Clinical evidence has recognized various etiological factors of ICH, including hypertension and infection. Extravasation of blood into brain parenchyma and formation of hematoma could lead to neuroinflammation and edema. After ICH induction, local brain tissue is promptly deformed, and the pathological processes of excitotoxicity, apoptosis and inflammation subsequently occurred [4]. Clinical reports have shown that the progressive conditions of patients deteriorate through the inflammatory cascade and secondary edema formation in ICH-induced brain injury [5].

Brain inflammation appears to be a key factor of secondary cerebral injuries, because ICH can induce prominent and complicated inflammatory responses, including resident microglial, astrocytic activation and circulating inflammatory leukocyte infiltration [6]. Activated microglial cells contributed to the release of various inflammatory mediators and cytokines and led to tissue injury after ICH [7]. Therefore, the approaches of anti-inflammatory and anti-matrix metalloproteinase (MMP)-9 therapies may be regarded as improving the outcome of ICH and strokes [8,9]. Hemorrhagic strokes are life-threatening, because existing therapies have limited effect [10]. Hence, it is desirable to provide new and effective anti-ICH drugs to patients more expeditiously [11].

The fruiting bodies of *Antrodia camphorata* (AC) have been used as a traditional Chinese medicine for the treatment of hypertension, hepatitis and cancers [12]. Recently, it has been reported that these fruiting bodies and their fermented culture broth exerted hepatoprotective [13], anti-inflammatory [14], and anti-amyloid β-protein-induced neurodegenerative effects [15]. It also proposed that AC could be considered as a potential therapeutic agent in neurodegenerative treatment [16]. In particular, ergosta-7,9(11),22-trien-3β-ol (EK100, Figure 1A), which is a bioactive and principal constituent that was isolated from the AC submerged whole broth, possesses a strong anti-analgesic and anti-inflammatory effects in mice models with a range of doses (5 and 10 mg/kg Body Weight) [17]. Additionally, this herbal component EK100 exerts anti-diabetic and anti-photodamaging effects in vivo [18,19]. Exclusively, the causes of ischemic stroke brain injuries were improved by EK100 [20]. This finding has inspired us to further evaluate the anti-neuroinflammatory effects of EK100 on microglial activation and its possible anti-neuroinflammatory mechanism in a collagenase-induced ICH mouse model.

## 2. Results

### 2.1. Effects of Ek100 on Neurobehavioral Deficits and T2 Images of Brain Injurious Lesion After Intracerebral Hemorrhage (ICH)

The neurological studies of manipulated mice were assessed at 24 h after ICH. For the vehicle-treated mice without ICH (NS-intrastriatal-microinjected group, sham control), the neurobehavioral score was 17.6 ± 0.5 (*n* = 8) with a median score of 18 (Figure 1B). After ICH induction, the neurobehavioral deficits were markedly reduced at a score of 9.8 ± 2.1 (*n* = 8) with a median score of 10 (Figure 1B). Notably, the neurological deficits were significantly improved by post-treatment of EK100 (30 mg/kg), as the score was 13.3 ± 1.9 (*n* = 8) with a median neurological score of 14 (*n* = 8) compared with the vehicle-treated ICH group (*p* < 0.001, Figure 1B). Furthermore, mice had T2-weighted MRI at 24 h after ICH. MRI showed the enlargement of an ipsilateral injurious lesion induced by ICH (Figure 1C). There were hyper-intense areas within the ipsilateral lesions, which showed the signs of hemorrhage and surrounding edema with low signal. Treatment of EK100 (30 mg/kg) reduced ICH-induced brain injury of enlargement lesions as shown in MRI imaging. As shown in Figure 1D, the lesion index was markedly increased by ICH induction by 64.7 ± 17.1 compared with 4.9 ± 5.8 of the non-ICH groups (sham, *n* = 4). After treatment of EK100 (30 mg/kg), the lesion index was significantly reduced at 38.1 ± 16.8 (*n* = 4) (*p* < 0.05, Figure 1D).

### 2.2. Effects of EK100 on the Expression of Cerebral COX-2/iNOS Proteins after ICH

To evaluate whether EK100 downregulates the neuroinflammatory mediators, we analyzed the expression of COX-2 and iNOS proteins’ expression in the manipulated brain tissues. A substantial increase in COX-2 or iNOS expression was revealed in the ipsilateral lobe compared with the contralateral lobe at 24 h after ICH (*p* < 0.001, Figure 2A,B). The post-ICH oral treatment with EK100 (30 mg/kg) significantly reduced COX-2 expression in the ipsilateral tissue (*p* < 0.05, Figure 2A). However, ICH-induced iNOS expression was not attenuated by EK100 treatment at the exact dosage (Figure 2B). These findings suggest that EK100 has preferentially effects to blockade COX-2 expression in ipsilateral lesions.

### 2.3. Effects of EK100 on the Brain Gelatinolytic Activity of MMP-9 after ICH In Vivo and on Collagenase VII-mediated Gelatinolysis In Vitro

To elucidate the protective effects of EK100 on ICH-induced brain tissue distortion, the striatal MMP-mediated gelatinolysis was identified by reference to its respective standard from a human fibrosarcoma cell line (HT1080) in the zymographic analyses. MMP-9-mediated gelatinolysis significantly and robustly increased in the ipsilateral brain tissues compared with the contralateral brain tissues at 24 h after ICH (*p* < 0.001, *n =* 5, Figure 3A). The administration of EK100 (30 mg/kg) markedly attenuated the induction of MMP-9-mediated gelatinolysis in the ipsilateral brain tissues (*p* < 0.05, *n =* 5, Figure 3A).

To investigate whether the anti-ICH effects of EK100 is mediated through downregulation of inflammatory mediators or inhibition of collagenase VII activity, we evaluated the in vitro enzyme activity of collagenase VII using zymographic methods. The high concentration of EK100 (50 μM) did not affect the degradative activity of collagenase VII (0.02 and 0.04 U) as compared with the vehicle-treated group (*n =* 3, Figure 3B). The positive control EDTA (2 mM) was shown to strongly inhibit the degrading activity of collagenase VII (*n =* 3, Figure 3B). These results support the notion that EK100 is not a collagenase VII inhibitor, but rather an anti-neuroinflammatory protectant after collagenase VII-induced ICH.

### 2.4. Effects of EK100 on the Expression of COX-2 Protein in Activated Microglial BV-2 cells, Primary Microglia, Cellular Viability and Productions of PGE_2_ and ROS

As shown in Figure 4A, LPS (150 ng/mL) induced the expression of COX-2 protein by seven-fold as compared with that of the control group (resting) in BV-2 cells. We pretreated BV-2 cells with various concentrations of EK100 (0.5, 1, 5, and 10 μM) for 15 min before stimulation with LPS. We found that EK100 attenuated the induction of COX-2 by LPS in a concentration-dependent manner (Figure 4A). On the other hand, the MTT assay showed that neither EK100 alone (0.5, 1, 5, and 10 μM) nor the solvent control (0.2% ethanol) significantly affected BV-2 cell viability (Appendix A). Notably, we also found that EK100 (10 μM) strongly inhibited LPS-induced COX-2 production in rat primary microglial cells. As shown in Figure 4B, LPS (150 ng/mL) induced COX-2 expression about 14-fold as compared with that of the control group in rat primary microglial cells. Similarly, EK100 (1 and 10 μM) attenuated the induction of COX-2 by LPS in a concentration-dependent manner (Figure 4B). Consistently, PGE_2_ level was enormously increased to 24.5 ± 4.1 pg per 1 *×* 10^6^ cells after LPS treatment as compared to the resting condition (1.3 ± 1.1 pg per 1 *×* 10^6^ cells). EK100 markedly inhibited PGE_2_ production to 19.0 ± 5.1, 16.7 ± 3.2, and 9.5 ± 2.3 pg per 1 *×* 10^6^ cells at the concentrations of 1, 5 and 10 μM, respectively (*n =* 3, Figure 4C). On the other hand, the LPS-induced microglial ROS generation was quantified using DCFH-DA probe by flow cytometry. As shown in Figure 4D, LPS exposure significantly increased microglial ROS production. EK100 markedly attenuated intracellular ROS production in a concentration-dependent manner in BV-2 cells (Figure 4D).

### 2.5. Effects of EK100 on LPS- and Thrombin-Induced MMP-9 Gelatinolysis in Astrocytes

In our preliminary studies, we found that LPS (100, 200, 500 and 1000 ng/mL) induced 3.0 ± 0.6-, 3.8 ± 0.4-, 4.3 ± 0.7-, and 4.6 ± 1.1-fold, respectively, of MMP-9-mediated gelatinolysis in the culture medium of primary astrocytes as compared with the resting condition (*n* = 3, data not shown). Interestingly, after pretreatment of astrocytes with EK100 (1, 5, and 10 μM) for 15 min followed by the addition of LPS (500 ng/mL), we found that EK100 caused concentration-dependent attenuation of MMP-9-mediated gelatinolysis. As shown in Figure 5A, LPS-induced MMP-9-mediated gelatinolysis was inhibited by EK100 (1, 5, and 10 μM) by 4.9 ± 1.0-, 3.8 ± 0.7-, and 3.0 ± 0.5-fold compared with 5.4 ± 0.8-fold of the vehicle group under LPS stimulation (*n* = 4), respectively. Furthermore, thrombin (10 U/mL)-induced MMP-9-mediated gelatinolysis was markedly attenuated by EK100 (1, 5, and 10 μM) by 6.0 ± 0.9-, 5.1 ± 0.6-, and 3.3 ± 1.3-fold compared with 5.5 ± 0.6-fold of the vehicle group under thrombin stimulation (*n* = 3), respectively (Figure 5B).

### 2.6. Effects of EK100 on IκBα Degradation in Activated Microglial BV-2 Cells

To investigate the inhibitory mechanisms of EK100 on the reduction in COX-2 expression, we studied several signaling molecules, including NF-κB and MAPK pathways. First, immunoblotting analyses revealed that LPS treatment (150 ng/mL) caused a rapid and time-dependent degradation of IκBα in immunoreactive bands (Figure 6A, lanes 2–5). It was shown that the IκBα protein was significantly degraded within 45 min after LPS stimulation and gradually returned after 60 min in BV-2 cells. However, EK100 (0.5–10 μM) could not reverse the degradation of IκBα with 45 min of LPS stimulation (Figure 6B, lanes 3–6). These results implied that EK100 did not affect the activation of NF-κB signaling.

### 2.7. Effects of EK100 on ERK, p38, and JNK MAPK Activation in Activated Microglial BV-2 Cells

To clarify the MAPK signaling pathways, the stimulation of BV-2 cells with LPS (150 ng/mL) at various times (15–90 min) resulted in time-dependent phosphorylation of p42/44 ERK (data not shown). The peak activation by LPS occurred after 30 min of stimulation and returned to basal levels after 60 min. After EK100 pretreatment (0.5–10 μM), which was followed by stimulation with LPS for 30 min, we found that EK100 partially affected the activation of p42/44 ERK MAPK (Figure 7A).

Next, the effect of EK100 on LPS-induced p38 MAPK activation was evaluated. The stimulation of BV-2 cells with LPS (150 ng/mL) at various times (15–120 min) resulted in time-dependent phosphorylation of p38 MAPK (data not shown). The peak activation of p38 MAPK occurred after 30 min of stimulation by LPS and returned to basal levels after 120 min. After EK100 pretreatment (0.5–10 μM), which was followed by stimulation with LPS for 30 min, the activation of p38 MAPK was not affected by EK100 (Figure 7B).

Furthermore, we determined the effect of EK100 on LPS-induced JNK MAPK activation. As shown in Figure 7C, the LPS (150 ng/mL)-induced phosphorylation of JNK MAPK (54 kDa) increased by up to 5.5 ± 0.6-fold compared with the resting control at 45 min after stimulation. Pretreatment with various concentrations of EK100 for 15 min before LPS indicated that EK100 (0.5, 1, 5, and 10 μM) cause concentration-dependent attenuation of LPS-induced JNK phosphorylation in BV-2 cells. EK100 (0.5, 1, 5, and 10 μM) inhibited JNK phosphorylation by 5.5 ± 1.9-, 3.9 ± 0.4-, 3.5 ± 1.2- and 2.4 ± 0.5-fold, respectively (Figure 7C).

### 2.8. Effects of EK100 on JNK MAPK Activation in the Ipsilateral Brain after ICH In Vivo

According to Western blotting analyses of ipsilateral brain tissues, the vehicle-treated ICH mice had significantly higher expression levels of p-JNK and p-ERK MAPKs than the control sham mice given vehicle (NS, Figure 8A,B). Under ICH conditions, EK100 post-treatment (30 mg/kg) markedly reduced p-JNK expression levels as compared with the vehicle-treated ICH mice (Figure 8A). However, EK100 exerted no effect on the ipsilateral ERK activation (Figure 8B). These in vivo results of ICH on JNK inhibition were consistent with our in vitro data showing the effects of EK100 on JNK signaling in BV-2 microglial cells.

## 3. Discussion

The present study demonstrates that the natural compound EK100 from *Antrodia camphorata* possesses the neuroprotective effects after collagenase-induced ICH and anti-microglial activation. The bacterial collagenase-injected model is commonly used to study the ICH-induced neuroinflammatory response and neurotoxicity [21]. Although there were some suggestions and pitfalls in this collagenase model [22], this model has still been widely used in the preclinical studies of strategies for ICH treatment [21]. On the other hand, the animal model of vascular perforation seems to mimic human brain hemorrhage conditions [23]. The blood constituents from disrupted vessels participate in hematoma-mediated striatal tissue injury [24]. Although there are some inherent differences between experimental and clinical ICH [25], this model is widely used to approach and clarify various neuroprotectants for ICH therapeutics. Our in vivo findings indicated that collagenase-induced ICH pathologically leads to increases in both COX-2 protein expression and MMP-9 gelatinolysis. These observations may represent the response of brain cells suffering from hematoma-mediated insults. Among these pathological mediators, COX-2 and MMP-9 may also contribute to brain edema and neuronal dysfunction [26,27].

The elevated expression of the COX-2 protein, which is responsible for prostaglandin production, is presumably induced in different brain cells by thrombin and mechanical stress [28]. Thus, due to the activation of specific prostaglandin receptors, its downstream signaling could promote neuronal injuries following acute excitotoxicity, hypoxia, and stress induced by ICH [29]. The increase in COX-2 expression may contribute to the neuroinflammatory cascade because the COX-2 blocker has been shown to improve brain injury after ICH [30]. Our results confirmed that post-treatment of EK100 significantly reduced brain injury as shown in MRI evaluation following ICH-induced COX-2 expression in ipsilateral lobe and microglia cells in vitro. As the intact degrading activity of collagenase VII was not significantly affected by EK100, these findings implied that the anti-ICH function of EK100 might not act through the inhibition of collagenase-mediated vessel disruption. Consistently, it was reported that EK100 could markedly reduce COX-2 expression and exert anti-inflammatory functions in carrageenin-treated mice [17]. Until now, comprehensive pharmacokinetics and tissue distribution studies of EK100 have not been reported. However, a brain-related function of the ischemic stroke has been carried out [6]. Therefore, it is necessary to further investigate the systemic plasma and brain levels of EK100.

The increased MMP-9 level is an independent risk factor for poor prognosis of acute cerebral hemorrhage patients [31]. Much evidence strongly indicated that MMP participates in ICH-induced brain injury and blood–brain barrier disruption [10]. For example, it was found that astrocytic induction of MMP-9 and edema in brain hemorrhage [32]. In particular, a thrombin-induced cellular injury may be partially mediated by MMP-9 activation [33]. Furthermore, intranuclear MMPs also promote DNA damage and apoptosis in the neuronal ischemia/reperfusion condition [34]. Moreover, MMP-9 may play a deleterious role in acute brain injury during the early stage of ICH and ischemic stroke; therefore, the inhibition of MMP activity during the critical period may improve brain injuries [9,35]. In the preliminary studies, we found that EK100 exhibited the inhibition of astrocytic glial fibrillary acidic protein over-expression in the ipsilateral region of ICH (data not shown). Moreover, EK100 exerted potent inhibition on the MMP-9 activation in ICH-injured brain tissues in vivo and activated astrocytes in vitro.

In addition, iNOS expression may contribute to secondary brain injuries [36], because NO overproduction could interact with superoxide anion to produce toxic peroxynitrite [37] and activate MMP-9 [38]. The manipulation of microglial iNOS inhibition showed a reduction in cerebral damage after ICH insult [39]. However, the elevation of cerebral iNOS was not affected by EK100 in our experimental condition (24 h-ICH). According to this differential effect of EK100 on cerebral COX-2 and iNOS expression, it revealed that the major action of EK100 might not occur through the repression of hematoma formation in vivo. In the present study, we showed that increased COX-2 expression and MMP-9-mediated gelatinolysis occurred in the same period after ICH and that these conditions can be significantly attenuated by post-treatment with EK100. These findings suggested that EK100 has anti-neuroinflammatory actions against brain injury after ICH induction.

Microglial cells contribute to the amplification of ICH-mediated inflammatory injury [6]. The resident microglia also plays a crucial role in the development of neuronal death [40]. Activated microglial cells could produce inflammatory cytokines and COX-2, which are involved in neuroinflammation in both the hematoma site and in peri-hematomal regions after ICH [41]. Increased COX-2 expression is found in microglia, astrocytes and vascular cells after cerebral insults. Our results showed that EK100 significantly inhibited COX-2 expression in activated microglial BV-2 and primary cells. Consistently, LPS-induced PGE_2_ production of BV-2 cells was attenuated by EK100. In contrast, the inhibitory activity of EK100 on microglial COX-2 production was not due to its cytotoxic effect because under these concentrations, there was no significant difference in the cellular viability. These results of microglial COX-2 inhibition in vitro were also consistent with the findings of in vivo studies. Furthermore, it was also shown that EK100 (30 mg/kg) treatment significantly reduced the immunofluorescent numbers of reactive microglia as co-localization of Iba-1 and COX-2 in situ in the peri-ICH area at 24 h after ICH (Appendix A). According to these in vivo findings, it consistently and strongly supported the neuroprotective mechanisms of EK100 may involve, at least partly, the inhibition of microglial COX-2 expression.

The microglial activation through the Toll-like receptor (TLR)-4 signaling pathway has been reported to play an essential role in ICH-induced brain injury [42]. The ICH-induced erythrocyte-derived heme or HMGB1 act as an activator of TLR-4 and cause microglial activation and neuroinflammation [21]. TLR-4 signaling inhibition was protective and significantly reduced neurological deficits in ICH [43]. LPS could strongly activate TLR-4 signaling and induce microglia activation and neuroinflammation [44]. In general, COX-2 expression in macrophages or microglia was in response to stimulation by LPS and thrombin [45,46]. Therefore, the TLR-4 activator LPS was used as the microglial stimulator in anti-neuroinflammatory ICH studies [47,48]. It is well-known that LPS stimulates TLR-4, which thereafter activates NF-κB or MAPK pathways and triggers the expression of microglial COX-2 [49]. Consistently, an early increase in NF-κB activation and COX-2 expression occurred in brain lesions after the onset of ICH [50]. Therefore, the manipulation with NF-κB inhibitors markedly attenuated LPS-induced COX-2 expression in microglial cells [51]. A previous report has shown that LPS would induce IκB kinase (IKK) activation and IκBα degradation in BV-2 cells [52]. However, we showed that EK100 has no significant effect on the LPS-mediated degradation of IκBα in BV-2 cells. Interestingly, EK100 was found to downregulate NF-κB expression in ischemic stroke [20]. The other possible interaction of NF-κB by EK100 in ICH needs to be further explored. On the other hand, our results were consistent with the findings in the literature [49,53], which indicating that ERK1/2, p38, or JNK/SAPK were involved in LPS-induced signal transduction of COX-2 expression in macrophages or BV-2 cells. Additionally, JNK MAPK plays a crucial role in inflammation because this pathway is involved in microglial activation [54]. The suppression of JNK MAPK pathways significantly resulted in the downregulation of microglial COX-2 expression [55,56]. According to our findings, LPS-induced microglial ROS production was concentration-dependently attenuated by EK100. It was consistent that JNK pathways are involved in the ROS productivity in LPS-activated BV-2 microglia [57]. Moreover, it was suggested that the inhibition of JNK MAPK was a reasonable target to improve brain edema and the functional impairment of ICH [58] and ischemic stroke [59]. Additionally, the neuroprotective effects of minimally invasive hematoma aspiration might be associated with suppressing JNK activation in ICH [60]. Our findings revealed that EK100 significantly inhibited LPS-mediated JNK MAPK activation, rather than ERK or p38 MAPK activation, in microglial BV-2 cells. Consistently, the direct evidence was strongly clarified ipsilateral JNK inhibition by EK100 under ICH in vivo. However, it is thought that further investigation into the mechanism of EK100 on the JNK signaling target is needed.

In conclusion, the crucial findings of this study suggest that the neuroprotective effects of post-treatment of EK100 on ipsilateral injuries in collagenase-induced ICH mice are assumed to act through the inhibition of neuroinflammatory COX-2 expression and MMP-9 activation. According to the functional studies, EK100 may improve brain edema, neurobehavioral deficits, and MRI-related brain lesion. EK100 also exerts significant anti-microglial function through the downregulation of JNK MAPK activation and inhibition of COX-2 protein expression in vitro and in vivo. Further studies are warranted to clarify the JNK-upstream target of EK100, such as MKK4/7 or ASK1. Moreover, EK100 could inhibit microglial and astrocytic activation, the usage of EK100 is not limited to one factor; instead, it involves some mechanisms, most of which may be interrelated. From these points, it can be concluded that this naturally occurring herbal compound from *Antrodia camphorata* is a potential therapeutic agent in treating ICH, and translation evaluation by the clinical trials is warranted.

## 4. Materials and Methods

### 4.1. Chemicals and Reagents

Freeze-dried AC powder was provided by the Biotechnology Center of Grape King Inc., Chung-Li City, Taiwan. Ergosta-7,9(11),22-trien-3β-ol (EK100) was isolated from freeze-dried AC powder by Prof. Yueh-Hsiung Kuo. Briefly, the dry AC powder was purified with SiO_2_ chromatography. EK100 was eluted with 10% EtOAc in hexane, and then recrystallized with acetone. The HPLC profile exhibited a purity of over 99.9% [17]. In each in vitro experiment, ethanol (≥99.8 %, Sigma-Aldrich, Saint Louis, MO, USA) as a vehicle (V), was used at a constant final concentration of 0.2% (*v*/*v*). Collagenase (VII-S), phenylmethylsulfonyl fluoride (PMSF), aprotinin, brilliant blue G-colloidal concentrate, Brij 35 detergent solution, β-mercaptoethanol, sodium dodecylsulfate (SDS), dithiothreitol (DTT), *N*-(2-hydroxyethyl) piperazine-*N*′-(2-ethanesulfonic acid) (HEPES), carboxymethyl cellulose (CMC), leupeptin, lipopolysaccharide (LPS; Escherichia coli, serotype 0127: B8) and 3-(4,5-dimethylthiazol-2-yl)-2,5-diphenyl-tetrazolium bromide (MTT) were purchased from Sigma-Aldrich (St. Louis, MO, USA). Poly-l-lysine was obtained from Millipore Corporation (Billerica, MA, USA). COX-2 (mouse) polyclonal antibody (pAb) was purchased from Cayman Chemical (Ann Arbor, MI, USA). Horseradish peroxidase-conjugated anti-mouse or anti-rabbit IgG and the Western blotting detection system (ECL+ plus) were purchased from Amersham Biosciences (Sunnyvale, CA, USA). All other chemicals that were used in this research were of reagent grade.

### 4.2. Animal Manipulation and ICH Induction

Male mice (C57B2/6J) (23–28 g) were used in this study. All experiments and animal care were performed according to the Guide for the Care and Use of Laboratory Animals (National Academy Press, Washington, DC, USA, 1996). The experimental procedures were approved by the Taipei Medical University, Institutional Animal Care and Use Committee (LAC-2015-0195 and LAC-98-0112). After animals were anesthetized by chloral hydrate (400 mg/kg) [61], the collagenase-induced ICH mouse model was created with minor modifications, as previously described [62]. Briefly, following the manipulation of intrastriatal microinjection [63], a 26-gauge needle on a Hamilton syringe (10 μL-GASTIGHT, #1701; Reno, NV, USA) was advanced stereotactically through the burr hole into the right striatum (coordinates: 0.3 mm anterior, 2 mm lateral to the bregma, 3.8 mm ventral). Collagenase (type VII-S, 0.02 U in 1 μL saline) was infused into the area of the caudate/putamen at a rate of 0.2 μL/min over 5 min using a microinfusion pump (Singa Co., Taoyuan City, Taiwan). After an additional period of 5 min, the needle was withdrawn slowly. Then, the burr hole was occluded with bone wax, and the incision was closed. The mice were allowed to recover with a controlled heating lamp and pad (37 °C). Sham animals received only needle insertion with an identical volume of normal saline. EK100 was suspended in a vehicle carboxymethyl cellulose (CMC, 0.5% *w*/*v*). At 30 min after collagenase injection, EK100 (30 mg/kg Body Weight, BW) was post-ICH orally administered to the mice. The volume of oral administration was controlled as 100 μL/20 g BW mouse. The control group of animals received an identical administration of the vehicle.

Following the previously described method [63], 24 h-manipulated animals were anesthetized and perfused transcardially with 4 °C normal saline for 5–10 min till hepatic decongestion. The fresh brains were quickly removed and placed in ice-cold normal saline for 5 min. The brain samples were then sectioned coronally into four sequential parts from the frontal to the occipital lobe using an Adult Mouse Brain Matrix (Model 0530, Vibratome, St. Louis, MO, USA). The part with the injected site of the right and left hemispheres (3 mm-thick) were separately collected, snap-frozen in liquid nitrogen, and stored at −70 °C. Each frozen brain tissue was finely minced with sharp chilled scissors and then homogenized with a polytron 1 mL-homogenizer (Wheaton, IL, USA) by hand in 200 μL of ice-cold 50 mM Tris-HCl (pH 7.4) buffer, which contained 150 mM NaCl, 0.2% Triton X-100 and complete EDTA-free protease inhibitor cocktail tablet solution (1×) (Roche, Mannheim, Germany). The homogenates were centrifuged at high speed (13,000 G) for 30 min. The resulting supernatants were stored at −70 °C. Samples of the brain supernatant were normalized for protein concentration determination. In total, 10 or 50 μg of protein from each brain supernatant was applied to each lane for gelatin-substrate zymography or Western blotting, respectively.

### 4.3. Neurobehavioral Deficits

At 24 h after ICH was induced by the intrastriatal injection of collagenase, mice were evaluated and scored blindly for neurobehavioral deficits using the neurological scoring system [63]. The neurobehavioral study was composed of six tests, which included spontaneous activity, symmetry in the movement of four limbs, forepaw outstretching, climbing, body proprioception and response to vibrissae touch. The score given to each mouse after the evaluation was the summation of all six individual test scores. The minimum neurological score was 3, and the maximum score was 18.

### 4.4. Magnetic Resonance Imaging (MRI) and Brain Injury Measurement

Mice were anesthetized with 2% isoflurane/air mixture throughout MRI examination. MRI was performed in a 7-Tesla PharmaScan with a 160 mm horizontal bore (Bruker, Germany) at the Taipei Medical University (TMU), Translational Imaging Research Center. It was used a T2 fast spin-echo (repetition time/echo time = 2,500/33 ms) with a field of view of 16 × 16 mm^2^, matrix of 256 × 256 and 10 coronal slices (0.75-mm thick). The single image was preserved as 256 × 256 pixel picture for T2 lesion evaluation. The brain injury evaluation was based on 3rd to 7th sections whose center was the anterior commissure layer. The volumes of different brain regions were gated and determined by Image J (https://imagej.nih.gov/ij/, accessed on 1 April 2021; NIH, Bethesda, MD, USA). The volume of ipsilateral lesions was gated and determined by the image with the hyper-intense hemorrhage and surrounding hypo-intense penumbra areas. The injured value was calculated by [(volume of the contralateral hemisphere)—(volume of ipsilateral hemisphere—volume of ipsilateral lesion)]/(volume of the contralateral hemisphere) × 100% [64].

### 4.5. Substrate Embedded Zymography

The matrix metalloproteinase (MMP)-mediated gelatinolytic zymography was performed as described previously, with minor modifications [63]. Additionally, the anti-collagenase VII activity of EK100 in vitro was evaluated as previously described, with minor modifications [65]. The specific gelatinolytic bands were analyzed using an identical digital imaging system and analytical software as previously described.

### 4.6. Cell Cultivation

The mouse BV-2 microglial cells were grown in DMEM and supplemented with 10% heat-deactivated fetal bovine serum (FBS), penicillin G (100 units/mL)/streptomycin (100 μg/mL), l-glutamine (3.65 mM), HEPES (18 mM), and NaHCO_3_ (23.57 mM) in a humidified 37 °C incubator with 5% CO_2_. The cells were treated with vehicle or with indicated concentrations of EK100 for 15 min, and then treated with LPS for additional indicated times. After stimulation, the cells were disrupted using lysis buffer, finally centrifuged and collected as previously described [66].

Primary cultures of microglial cells and astrocytes were obtained from brain cortices of Wistar rats (7 day old) with some modification [67,68]. Briefly, the mixed glial cells were obtained and seeded in 75 cm^2^ flasks (No. 430641, Corning, NY, USA). Media were changed every 3 days. After incubation for 8, 11 and 14 days at 37 °C, cultures were shaken for 2 h at 120 rpm and 37 °C in an orbital shaker (No. 430641, Digisystem Lab. Instrument Inc., New Taipei City, Taiwan), the detached cells were collected as primary microglial cells as previously described [68]. Thereafter, the remained cultures were shaken stronger for 6 h at 200 rpm and 37 °C. Detached cells were washed out and the remaining astrocyte monolayer was trypsinized and replated in the flask coated with poly-l-lysine (50 μg/mL). After reaching confluence, astrocytes were trypsinized and replated onto the coated 12 well-culture plate (No. 3513, Corning, NY, USA) for each experiment at a density of 1 × 10^5^ cells/mL. According to the method of isolation, approximately 95% of the cells showed immunoreactivity (IR) to glial fibrillary acidic protein (GFAP). The cells were cultured in RPMI-1640 supplemented with 10% FBS for overnight. Then, the culture media were changed without FBS for 24 h. Primary microglial cells or astrocytes were preincubated with EK100 for 15 min followed by either addition of LPS or thrombin for 24 h. The conditioned media were collected, centrifuged and stored at −70 °C for less than 2 weeks.

### 4.7. Cell Viability

The cell viability of BV-2 cells was evaluated after 24 h of continuous exposure to either vehicle or EK100 (0.5, 1, 5 or 10 μM) using a colorimetric assay, which was based on the ability of mitochondria to reduce the MTT [69]. The percentage of cell viability was calculated as the absorbance of treated cells/control cells × 100%.

### 4.8. Determination of the PGE_2_ Levels by the Enzyme-Linked Immunosorbent Assay (ELISA)

The levels of PGE_2_ production by BV-2 cells in the conditioned media were assayed by using the ELISA kit (BioLegend, San Diego, CA, USA). The quantitative levels of PGE_2_ were determined as previously instructed and described [69].

### 4.9. Detection of Intracellular ROS Levels

To examine where EK100 regulate LPS-induced ROS production, we measured intracellular ROS level in BV-2 cells using the redox-sensitive fluorescent dye DCFH-DA and flow cytometry. The BV-2 cells were pretreated with vehicle or EK100 (5, 10 and 20 μM) for 30 min, then stimulated with LPS (150 ng/mL) for 24 h. After stimulation, BV-2 cells were purchased and incubated with DCFH-DA for 40 min. Then, the cellular DCF fluorescent intensities were analyzed by flow cytometry (BD AccuriTM C6, Piscataway, NJ, USA). The quantitative levels of intracellular ROS were determined as previously described [69].

### 4.10. Western Blot Analyses

For the determination of the expression of COX-2, iNOS and phosphorylated JNK in brain tissues or COX-2, IκBα, and phosphorylated MAPKs (ERK, p38, and JNK) in microglial cells, Western blotting assay was performed as described previously [63]. BV-2 cells (2 × 10^5^/mL; 2.5 mL/well) were cultured in a 6-well plate as described previously. The cells were pretreated with EK100 (5–50 μM) for 15 min, which was followed by the addition of LPS (150 ng/mL) for the indicated times, and incubated at 37 °C. After incubation and gentle washing, the cells were broken using ice-cold lysis buffer (50 mM HEPES, 50 mM NaCl, 5 mM EDTA, 1% Triton X-100, 2 mM DTT, 1 mM PMSF, 10 μg/mL aprotinin, and 10 μg/mL leupeptin, pH 7.0) for 30 min. Additionally, phosphatase inhibitors (10 mM sodium fluoride, 1 mM sodium orthovanadate, and 5 mM sodium pyrophosphate) were added to the lysis buffer for the phosphorylated MAPK analysis. The cellular lysates were harvested by centrifugation (12,000 rpm, 20 min). After Western blot manipulation, the blots were incubated with specific antibodies against COX-2, IκBα, and phosphorylated MAPKs, or α-tubulin. The data for specific protein levels are relative multiples in relation to the control. Quantitative densitometry analyses were performed as previously described.

### 4.11. Statistical Analyses

Data analyses were performed using the software SigmaStat 3.5 (SYSTAT Software, San Jose, CA, USA). The experimental results are expressed as the mean ± SD and are accompanied by the number of observations. The statistical analysis was performed using a one-way ANOVA. The data were assessed using Student–Newman–Keuls test. The neurodeficit scores were expressed as medians. A *p* value of less than 0.05 was considered statistically significant.

## Figures and Tables

**Figure 1 molecules-26-02970-f001:**
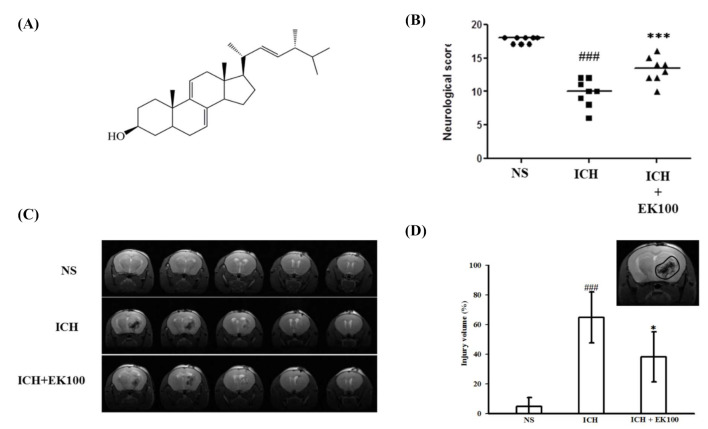
Effects of ergosta-7,9(11),22-trien-3β-ol (EK100) on neurobehavioral deficits and neuroimaging abnormalities after intracerebral hemorrhage (ICH) in vivo. C57BL/6 mice were post-ICH orally administered EK100 (30 mg/kg) or vehicle at 30 min after the intrastriatal injection of 0.02 U collagenase to induce ICH. (**A**) The chemical structure of EK100 was shown. (**B**) The neurobehavioral deficit evaluation scores were shown as all six individual test scores after 24 h (*n* = 8). The evaluations were described as details in the Methods section. (**C**) T2-weighted MRIs at 24 h after ICH treated with vehicle or EK100 (30 mg/kg) and their non-ICH control. (**D**) Bar graph demonstrating the lesion index expressed, as detailed in Materials and Methods. Values were expressed as the mean ± SD (*n* = 4). ^###^
*p* < 0.001 compared with ipsilateral NS (normal saline)-intrastriatal injected with vehicle treatment group; * *p* < 0.05, *** *p* < 0.001 compared with ipsilateral collagenase-intrastriatal injected with vehicle treatment group (ICH). The gated-locus used for evaluation of brain injury was illustrated.

**Figure 2 molecules-26-02970-f002:**
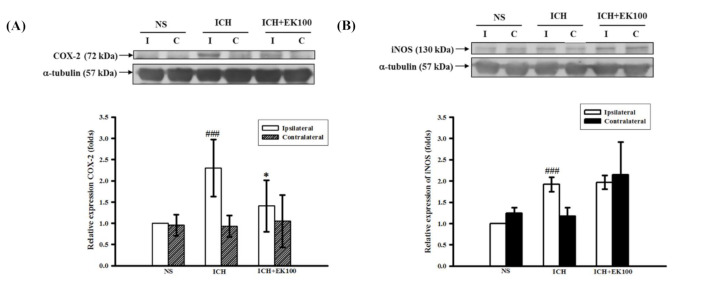
Effects of EK100 on the expression of brain COX-2 and iNOS proteins after ICH. C57BL/6 mice were post-ICH orally administered EK100 (30 mg/kg) at 30 min after the intrastriatal injection of 0.02 U collagenase to induce ICH. Brain slices (3 mm) were obtained at 24 h after ICH, and their ipsilateral (I) and contralateral (C) regions were homogenated and analyzed for the expression of COX-2 (**A**, *n =* 6) and iNOS (**B**, *n =* 5) proteins by Western blots. The relative changes (folds) were represented as the mean ± SD. ^###^
*p* < 0.001 compared with ipsilateral NS-intrastriatal injected with vehicle treatment group; * *p* < 0.05 compared with ipsilateral collagenase-intrastriatal injected with vehicle treatment group.

**Figure 3 molecules-26-02970-f003:**
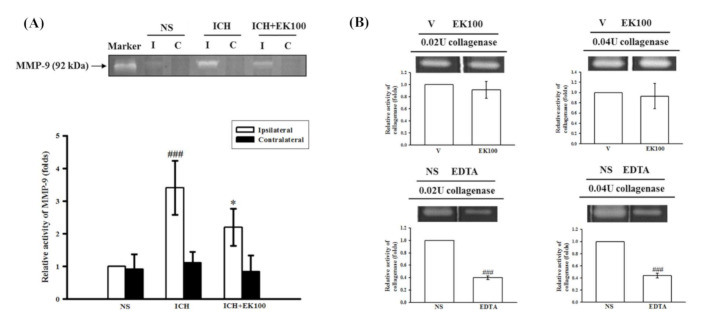
Effects of EK100 on the gelatinolytic activity of cerebral MMP-9 after ICH in vivo and collagenase VII enzyme in vitro. (**A**) C57BL/6 mice were administered EK100 with an oral dosage of 30 mg/kg at 30 min after the intrastriatal injection of 0.02 U collagenase to induce ICH. Brain slices (3 mm) were obtained at 24 h after ICH, and their brain tissue homogenates of ipsilateral (I) and contralateral (C) regions were evaluated for MMP-9 activity using zymography. The relative changes (folds) were represented as the mean ± SD (*n =* 5). ^###^
*p* < 0.001 compared with ipsilateral NS-intrastriatal injected with vehicle treatment group; * *p* < 0.05 compared with ipsilateral collagenase-intrastriatal injected with vehicle treatment group. The marker was represented as the positive control of MMP-9-mediated gelatinolysis from the medium of HT1080 cells. (**B**) The enzymatic units of collagenase VII (0.02 and 0.04 U) were used to evaluate the catalytic activity by gelatin zymography in vitro. EK100 (50 μM) and its vehicle (V) or the positive control (EDTA, 2 mM) and its vehicle (NS) were added to the reacting buffer with the manipulated gels as described in the Methods section. The relative changes (folds) were represented as the mean ± SD. ^###^
*p* < 0.001 compared with the vehicle-treated condition in vitro.

**Figure 4 molecules-26-02970-f004:**
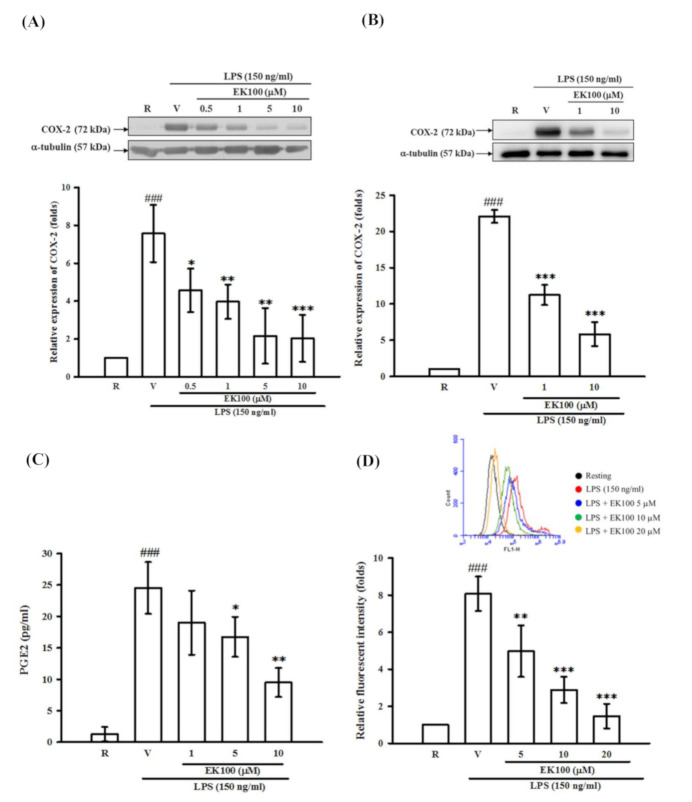
Effects of EK100 on the LPS-induced expression of the COX-2 protein in microglial BV-2 and primary microglial cells, and productions of PGE_2_ and ROS. (**A**) BV-2 cells (2 × 10^5^ cells/mL) were dispensed on 6-well plates and treated with different concentrations of EK100 (0.5, 1, 5 and 10 μM) or vehicle (V) for 15 min before stimulation with LPS (150 ng/mL) for 24 h. Then, the cell lysates were obtained and analyzed for COX-2 protein expression by Western blots. The relative changes (folds) were represented as the mean ± SD (*n =* 3). (**B**) Primary microglial cells were dispensed on 6-well plates and treated with different concentrations of EK100 (1 and 10 μM) or vehicle for 15 min before stimulation with LPS (150 ng/mL) for 24 h. The relative changes (folds) of COX-2 protein expression were represented as the mean ± SD (*n =* 4). (**C**) PGE_2_ contents in conditioned media of BV-2 cells were evaluated by ELISA. The values were analyzed and represented as the mean ± SD (*n =* 3). (**D**) The BV-2 cells were pretreated with vehicle or EK100 (5, 10 and 20 μM) for 30 min, then stimulated with LPS (150 ng/mL) for 24 h. After stimulation, BV-2 cells were purchased and incubated with DCFH-DA for 40 min. Then, the cellular DCF fluorescent intensities were analyzed by flow cytometry (BD AccuriTM C6, NJ, USA). The results were analyzed and represented as the mean ± SD (*n =* 3). ^###^
*p* < 0.001 compared with the resting group (R); * *p* < 0.05, ** *p* < 0.01, *** *p* < 0.001 compared with the vehicle under stimulation.

**Figure 5 molecules-26-02970-f005:**
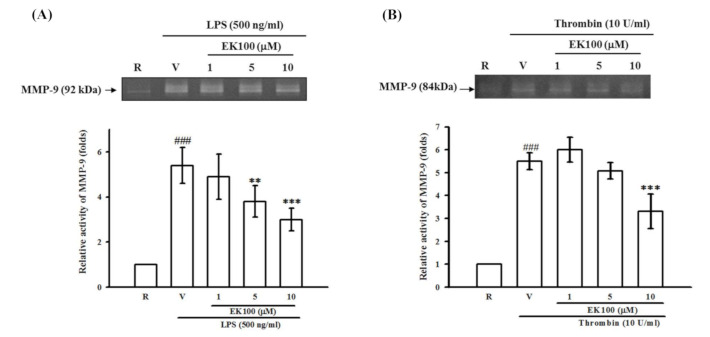
Effects of EK100 on LPS- and thrombin-induced MMP-9 activation in astrocytes. Rat primary astrocytes were pretreated with the vehicle (V) or EK100 (1, 5, and 10 μM), and stimulated with either (**A**) LPS (500 ng/mL) or (**B**) thrombin (10 U/mL) for 24 h as indicated. Cell-free supernatants were then assayed for MMP-9 activity by gelatin zymography. The relative multiples of densitometric data are shown as the mean ± SD (*n =* 3 and 4). ^###^
*p* < 0.001 compared with the resting group (R); * *p* < 0.05, ** *p* < 0.01 compared with the vehicle under stimulation.

**Figure 6 molecules-26-02970-f006:**
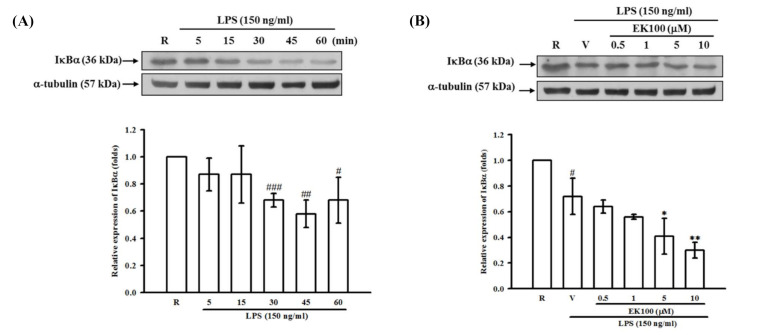
Effect of EK100 on the LPS-induced degradation of IκBα in BV-2 cells. Western blot analyses were demonstrated (**A**) the time course of LPS-induced IκBα degradation in BV-2 cells. The cells (2 × 10^5^ cells/mL) were dispensed on 6-well plates and stimulated with LPS (150 ng/mL) for 5, 15, 30, 45, and 60 min as indicated, respectively. The data were shown as the mean ± SD (*n =* 3). ^#^
*p* < 0.05, ^##^*p* < 0.01, ^###^
*p* < 0.001 compared with the resting group (R). (**B**) BV-2 cells (2 × 10^5^ cells/mL) were pretreated with EK100 (0.5, 1, 5 and 10 μM) or vehicle (V) for 15 min and stimulated with LPS (150 ng/mL) for 45 min. The cell lysates were obtained and analyzed with a specific IκBα antibody using Western blotting methods. The data were shown as the mean ± SD (*n =* 3). ^#^
*p* < 0.05 compared with the resting group (R); * *p* < 0.05, ** *p* < 0.01 compared with the vehicle under stimulation.

**Figure 7 molecules-26-02970-f007:**
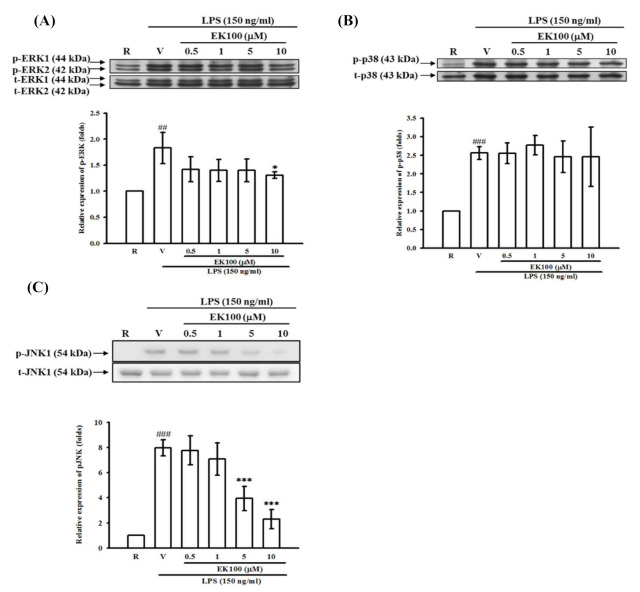
Effects of EK100 on LPS-induced MAPK activation in BV-2 cells. BV-2 cells (2 × 10^5^ cells/mL) were dispensed on 6-well plates and pretreated with EK100 (0.5, 1, 5, and 10 μM) or vehicle (V) for 15 min before stimulation with LPS (150 ng/mL) for the indicated time (30 min for ERK and p38; 45 min for JNK), respectively. The cell lysates were obtained and analyzed for phosphorylated ERK (1/2) (**A**), p38 (**B**), and JNK (**C**) MAPKs by Western blotting methods. The relative changes (folds) were represented as the mean ± SD (*n =* 4). ^##^
*p* < 0.01, ^###^
*p* < 0.001 compared with the resting group (R); * *p* < 0.05, ** *p* < 0.01 compared with the vehicle under stimulation.

**Figure 8 molecules-26-02970-f008:**
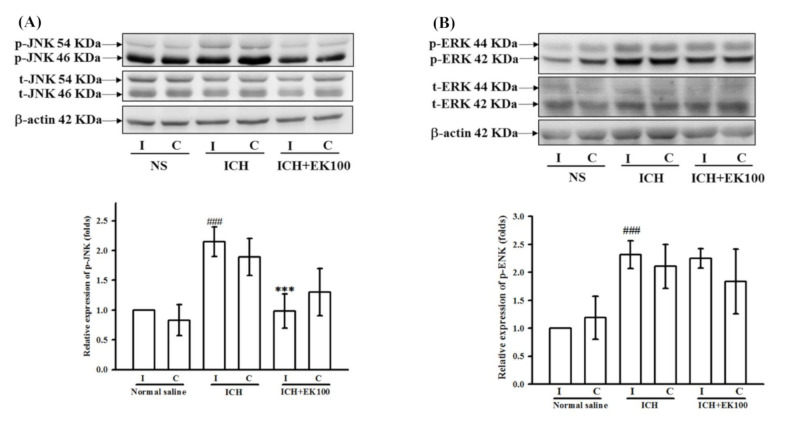
Effects of EK100 on the cerebral levels of p-JNK and p-ERK MAPK proteins after ICH. C57BL/6 mice were administered vehicle or EK100 with an oral dosage of 30 mg/kg at 30 min after the intrastriatal injection of 0.02 U collagenase to induce ICH. Brain slices (3 mm) were obtained at 24 h after ICH, and their ipsilateral (I) and contralateral (C) regions were analyzed for the expression of p-JNK (**A**, *n =* 4) and p-ERK (**B**, *n =* 4) proteins by Western blots. The relative changes (folds) were represented as the mean ± SD. ^###^
*p* < 0.001 compared with ipsilateral NS-intrastriatal injected with vehicle treatment group; *** *p* < 0.001 compared with ipsilateral collagenase-intrastriatal injected with vehicle treatment group (ICH).

## Data Availability

The data presented in this study are available on request from the corresponding author.

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
