# Peer review of "Ergosta-7,9(11),22-trien-3β-ol Alleviates Intracerebral Hemorrhage-Induced Brain Injury and BV-2 Microglial Activation"

_molecules, 2021, doi:10.3390/molecules26102970_

Round 1

Reviewer 1 Report

The authors in their article present research related to the activity of ergosta-7,9(11),22-trien-3β-ol (EK100) as a potential agent in intracerebral hemorrhage (ICH) reduction. The results of the study showed that EK100 improved neurobiological deficits and reduced brain injury lesions  in mice with induced ICH. EK100 attenuated COX-2 expression and MMP-9 activity in the ipsilateral region of the brain following ICH. This was confirmed in microglial cells. EK100 abrogated prostaglandin E2 and ROS generation. Furthermore, EK100 inhibited ipsilateral JNK activation.

Overall, the manuscript is well presented and adds novel information to the current knowledge regarding the bioactivity of EK100. I have the following questions for the authors:

  • In the results section regarding the effects of EK100 on neurobehavioral deficits some information requires clarification. I would suggest changing the information in line 89 from ‘After ICH treatment’ to ‘After ICH induction’ and provide information regarding the factor used for ICH induction. Furthermore, in line 89 the authors mentioned that neurobehavioral deficits were reduced, did the authors mean that ICH induction reduced the neurobiological score and worsened neurobehavioral deficits in mice.
  • The results of this study did not show the influence of EK100 on IκBα degradation, hence the authors concluded that EK100 does not influence NF-κB signaling. Previous studies on EK100 showed downregulation of NF-κB in ischemic stroke and since NF-kB expression has been shown in some instances to be influenced independent of IκBα, the authors could confirm the effects of EK100 on the level of NF-κB expression.
  • Some language corrections are required, examples of corrections are provided below:

Line 56: apoptosis and inflammation subsequently occurred

Line 65: therapies may be regarded as

Line 80: this finding has inspired us

Line 100: the lesion index was significantly reduced at

Line 386: and inhibition of COX-2 protein expression

Author Response

Responses to comments of reviewer # 1

We thank the reviewer for perusing the manuscript and for the constructive and appreciable comments. Per the Editor’s instructions, we would like to offer the following replies to the specific points raised by the reviewer. The changes have been made in the revised paper are highlighted in red color.

  1. We agree with the reviewer’s sensible comment that we have changed “After ICH treatment” to “After ICH induction” for better understanding (Line 89). Thank you for your kind suggestion.
  2. Thank you for your kind opinion. Yes, we did mean that ICH induction reduced the neurological score and worsened neurobehavioral activities in mice (Line 89, Fig. 1B of the new version).
  3. Thank you for your kind suggestion. According to the previous studies, the authors found the distribution of p65NF-kB (p65NF-kB staining) was downregulated by EK100 (please refer Fig. 4 and its legend of the original paper: p4727 and p4729, Food Funct., 2019, 10, 4725). It was proposed the infiltration of inflammatory leukocytes were inhibited.

However, we found the anti-microglial activation being not through NF-kB inhibition by decrease of IkBa degradation (original Fig. 6) or phosphorylation of p65 (the new experiments as shown below, n=3) by EK100. (please refer pdf file for its figure)

  1. Thank you for your kind suggestion. English language and grammar have been corrected by your opinions. The changes have been made in the revised paper (the red marked).
    • “apoptosis and inflammation were subsequently occurred” (Line 56).
    • “…therapies may be regarded.” (Line 65).
    • “This finding has inspired us…” (Line 80).
    • “..the lesion index was significantly reduced at…” (Line 100).
    • “..and inhibition of COX-2 protein expression…” (Line 566).

Reviewer 2 Report

In this paper the authors describe the potential of ergosta-7,9(11),22-trien-3β-ol (EK100), a bioactive ingredient from Asian medicinal herb, Antrodia camphorate, for treating intracerebral hemorrhage (ICH). ICH accounts for 15 to 30% of all strokes, and provokes an important number of patient deaths, so the development of new efficacious therapies is an important objective. In this context, the research topic is interesting and timeliness.

The results reported in this manuscript suggest that in vivo treatment with EK100 significantly attenuated neurobehavioral deficit and MRI-related brain lesion in the mice model of collagenase-induced ICH. Also, EK100 decreased inflammatory markers such as the expression of the inducible cyclooxygenase (COX)-2 and the activity of matrix metalloprotease (MMP)-9. In addition, EK100 inhibits JNK activation both in cellular systems and in vivo. The manuscript is clearly written and the methodology is technically sound. However, it is considered that some critical aspects need to be addressed before the work can be published to ensure that the conclusions are fully supported by the data.

Major concerns:

-The authors should provide experimental in vivo data about brain concentration of EK100 after oral administration of the compound.

-The authors should provide experimental evidences about the biological target of EK100.

Minor points:

-The structure of EK100 should be included in the manuscript.

Author Response

Responses to comments of reviewer # 2

We thank the reviewer for your constructive and appreciable comments. The changes have been made or highlighted in the red color in the revised manuscript.

  1. Thank you for your kind suggestion. Per your kind opinion, we have tried to conduct the brain PK of EK100 experiments by the methods of Zhao et al. (J. Chromatography B, 879: 1945-1953, 2011) in vivo. However, the time is really limited for the present condition. We had found some ergosterols (such as withanolide A: Drug Dev Res. 79: 339-351, 2018) and EK100 with the neuroprotective functions. It also proposed the permeability of the BBB being altered for drugs during neuroinflammation under strokes. However, the brain PK of EK100 need to be warranted by further studies. We are appreciated for your suggestions.
  2. Thank you for your kind suggestion. In fact, we had provided experimental evidences about the signal target JNK of EK100 in vitro and in vivo. However, the upstream targets of MKK4/7 or ASK1 need to be further investigated (Genes Dev. 15: 1419-1426, 2001; Transl Res. 2021 Apr 12;S1931-5244(21)00083-9). Therefore, we have added the new description of “Further studies are warranted to clarify the JNK-upstream target of EK100, such as MKK4/7 or ASK1” in the text of the revised manuscript (Line 567). It is appreciated for your kind suggestions.
  3. Thank you for your kind opinion. The structure of EK100 had been included in Fig. 1A of the manuscript in the new version.

Zhao YY, Cheng XL, Liu R, Ho CC, Wei F, Yan SH, Lin RC, Zhang Y, Sun WJ.Pharmacokinetics of ergosterol in rats using rapid resolution liquid chromatography-atmospheric pressure chemical ionization multi-stage tandem mass spectrometry and rapid resolution liquid chromatography/tandem mass spectrometry. J Chromatogr B Analyt Technol Biomed Life Sci. 2011, 879(21):1945-53.

Singh SK, Valicherla GR, Joshi P, Shahi S, Syed AA, Gupta AP, Hossain Z, Italiya K, Makadia V, Singh SK, Wahajuddin M, Gayen JR. Determination of permeability, plasma protein binding, blood partitioning, pharmacokinetics and tissue distribution of Withanolide A in rats: A neuroprotective steroidal lactone. Drug Dev Res. 2018,79(7):339-351.

Tournier C, Dong C, Turner TK, Jones SN, Flavell RA, Davis RJ. MKK7 is an essential component of the JNK signal transduction pathway activated by proinflammatory cytokines. Genes Dev. 2001, 15(11):1419-26.

Miller MR, Koch SR, Choi H, Lamb FS, Stark RJ. Apoptosis signal-regulating kinase 1 (ASK1) inhibition reduces endothelial cytokine production without improving permeability after Toll-like receptor 4 (TLR4) challenge. Transl Res. 2021, S1931-5244(21)00083-9. doi: 10.1016/j.trsl.2021.04.001. Online ahead of print.

Round 2

Reviewer 2 Report

My previous major concerns have not experimentally addressed. Therefore, at least the lack of these data (evidence of brain levels of EK100 and of the specific targets of the compound) should be mentioned in the text as limitations of the conclusions of the present study.

Author Response

Responses to comments of reviewer # 2

We thank the reviewer for your constructive and appreciable comments. The changes have been made or highlighted in the red color in the new revised version.

  1. Thank you for your kind suggestion. Per your kind opinion, we have mentioned in the text as limitations of the conclusions of the present study.

First, the limitations of the conclusions with the data of brain levels of EK100, the descriptions of “Till now, comprehensive pharmacokinetics and tissue distribution studies of EK100 have not been reported. However, a brain-related function of the ischemic stroke has been carried out [6]. Therefore, it is necessary to further investigate the systemic plasma and brain levels of EK100.” were added (Page 11, Line 7-11 in the new version).

Second, the limitations of the data of specific target of EK100, the descriptions of “However, it is thought that further investigation into the mechanism of EK100 on the JNK signaling target is needed.” were added, too (Page 12, Line 36-37).

We are appreciated for your kind suggestions.